# Pallidal Deep Brain Stimulation in DYT6 Dystonia: Clinical Outcome and Predictive Factors for Motor Improvement

**DOI:** 10.3390/jcm8122163

**Published:** 2019-12-06

**Authors:** Annika Danielsson, Miryam Carecchio, Laura Cif, Anne Koy, Jean-Pierre Lin, Göran Solders, Luigi Romito, Katja Lohmann, Barbara Garavaglia, Chiara Reale, Giovanna Zorzi, Nardo Nardocci, Philippe Coubes, Victoria Gonzalez, Agathe Roubertie, Gwenaelle Collod-Beroud, Göran Lind, Kristina Tedroff

**Affiliations:** 1Department of Women’s and Children’s Health, Karolinska Institutet, 17176 Stockholm, Sweden; kristina.tedroff@ki.se; 2Sachs’ Children and Youth Hospital, Stockholm South General Hospital, 11883 Stockholm, Sweden; 3Department of Pediatric Neuroscience, Fondazione IRCCS Istituto Neurologico Carlo Besta, 20131 Milan, Italy; mcarecchio@gmail.com (M.C.); giovanna.zorzi@istituto-besta.it (G.Z.); nardo.nardocci@istituto-besta.it (N.N.); 4Neurogenetics Unit, Fondazione IRCCS Istituto Neurologico Carlo Besta, 20126 Milan, Italy; 5Department of Neuroscience, University of Padua, 35128 Padua, Italy; 6Département de Neurochirurgie, Unité de Recherche sur les Comportements et Mouvements Anormaux, (URCMA), Centre hospitalier universitaire de Montpellier, 34090 Montpellier, France; a-cif@chu-montpellier.fr (L.C.); p-coubes@chu-montpellier.fr (P.C.); v-gonzalez@chu-montpellier.fr (V.G.); 7Faculty of Medicine, University of Cologne and Deparment of Pediatrics, University Hospital Cologne, 50924 Cologne, Germany; anne.koy@uk-koeln.de; 8Complex Motor Disorders Services, Evelina London Children’s Hospital, Children’s Neuromodulation, Children and Women’s Health Institute, King’s Health Partners, London SE1 7EH, UK; jean-pierre.lin@gstt.nhs.uk; 9Department of Clinical Neuroscience, Karolinska Institutet, 17177 Stockholm, Sweden; goran.solders@ki.se (G.S.); goran.lind@sll.se (G.L.); 10Department of Neurology, Karolinska University Hospital, 14186 Stockholm, Sweden; 11Department of Movement Disorders, Fondazione IRCCS Istituto Neurologico Carlo Besta, 20133 Milan, Italy; luigi.romito@istituto-besta.it; 12Institute of Neurogenetics, University of Luebeck, 23562 Luebeck, Germany; katja.lohmann@neuro.uni-luebeck.de; 13Medical Genetics and Neurogenetics Unit, Fondazione IRCCS Istituto Neurologico Carlo Besta, 20126 Milan, Italy; barbara.garavaglia@istituto-besta.it (B.G.); chiara.reale@istituto-besta.it (C.R.); 14Département de Neuropédiatrie, Centre hospitalier universitaire de Montpellier, 34295 Montpellier, France; a-roubertie@chu-montpellier.fr; 15INSERM U 1051, Institut des Neuroscience de Montpellier, 34091 Montpellier, France; 16Aix Marseille Univ, INSERM, MMG, 13385 Marseille, France; gwenaelle.collod-beroud@inserm.fr; 17Astrid Lindgren Children’s Hospital, Karolinska University Hospital, 17176 Stockholm, Sweden

**Keywords:** pallidal deep brain stimulation, DYT-THAP1 dystonia, Burke Fahn Marsden Dystonia Rating Scale, long-term follow-up

## Abstract

Pallidal deep brain stimulation is an established treatment in dystonia. Available data on the effect in DYT-THAP1 dystonia (also known as DYT6 dystonia) are scarce and long-term follow-up studies are lacking. In this retrospective, multicenter follow-up case series of medical records of such patients, the clinical outcome of pallidal deep brain stimulation in DYT-THAP1 dystonia, was evaluated. The Burke Fahn Marsden Dystonia Rating Scale served as an outcome measure. Nine females and 5 males were enrolled, with a median follow-up of 4 years and 10 months after implant. All benefited from surgery: dystonia severity was reduced by a median of 58% (IQR 31-62, *p* = 0.001) at last follow-up, as assessed by the Burke Fahn Marsden movement subscale. In the majority of individuals, there was no improvement of speech or swallowing, and overall, the effect was greater in the trunk and limbs as compared to the cranio-cervical and orolaryngeal regions. No correlation was found between disease duration before surgery, age at surgery, or preoperative disease burden and the outcome of deep brain stimulation. Device- and therapy-related side-effects were few. Accordingly, pallidal deep brain stimulation should be considered in clinically impairing and pharmaco-resistant DYT-THAP1 dystonia. The method is safe and effective, both short- and long-term.

## 1. Introduction

Dystonia is defined as “a movement disorder characterized by sustained or intermittent muscle contractions causing abnormal, often repetitive movements, postures, or both” [1]. It is often initiated or worsened by voluntary action. The clinical presentation, etiology, and pathophysiology are heterogeneous, ranging from adult-onset focal dystonia to pediatric-onset generalized dystonia, with sometimes life-threatening conditions. The dystonia may be isolated, combined with other movement disorders, or associated with other systemic or neurological symptoms. Some forms are acquired and others are inherited. Since the discovery of the *TOR1A* gene and its connection to DYT-TOR1A dystonia (also known as DYT1 dystonia) in 1997, more than 200 genes linked to dystonia have been identified, with a significant increase in recent years due to the development of next-generation sequencing techniques [2,3]. Dystonia often has a considerable negative impact on quality of life [4,5]. Oral treatment, such as baclofen and trihexyphenidyl, provides limited and transient improvement in some patients, and intramuscular injections of botulinum toxin A are applied mainly in cases of focal dystonias [6,7]. 

Deep brain stimulation (DBS) is a well-established, albeit invasive treatment option for dystonia, as well as other movement disorders, such as Parkinson´s disease and essential tremor [8,9,10,11,12,13]. Pallidal DBS, stimulation of the internal globus pallidus (GPi), has been evaluated for a range of different conditions, with dystonia of varying phenotype and etiology, and some patients with dystonia benefit more from this procedure than others [14]. With isolated generalized dystonia DBS produces good or sometimes even excellent results, with motor improvement ranging from 50% to 75% as assessed by a variety of methods [11,15]. On the other hand, the outcome of DBS in individuals with dystonia in combination with other neurological symptoms or an abnormal brain MRI is less predictable, often resulting in little or no improvement [16,17]. Thus, in order to select patients suitable for surgery, it has become of paramount importance to identify factors that predict improvement after DBS. The specific etiology of the dystonia is emerging as a relevant predictor of post-surgical improvement and the disease course after DBS [16,18,19]. A present challenge is to establish whether genetics can contribute to a better selection of the patients most suitable for surgery. It has been proposed that for rare disorders such as dystonia, systematic and multicenter efforts are needed to address genetic influences on DBS outcome [19].

DYT-THAP1 dystonia (also known as DYT6 dystonia) is an isolated dystonia caused by dominant mutations in the *THAP1* gene [20]. Onset typically occurs during childhood or adolescence [21]. Symptoms vary among affected individuals, but cranio-cervical, laryngeal, and oro-mandibular involvement are often observed and reported to be a very debilitating factor by children and their families. Generalization occurs in about 45% of mutation carriers [22]. The overall number of individuals with DYT-THAP1 dystonia who have undergone pallidal DBS to date is limited. The first published small case series, including 2–4 patients, reported only moderate responses to DBS [23,24,25]. Recently, however, a more favorable outcome, more similar to that observed in other isolated dystonias, has been reported [18,26].

We aimed to elucidate the clinical outcome of GPi-DBS on individuals with DYT-THAP1 dystonia and to identify potential factors that predict positive or negative changes in motor function.

## 2. Material and Method

This is a retrospective multicenter case series of individuals with DYT-THAP1 dystonia after pallidal deep brain stimulation. Patients were enrolled from 5 European centers (Stockholm, Sweden; Milan, Italy; Montpellier, France; London, England; and Cologne, Germany) and inclusion criteria were: patients affected by dystonia, with mutations/sequence variants in *THAP1*, who had undergone GPi-DBS with post-operative follow-up for at least 6 months at the time for data collection (2017–2018). Data were extracted from medical records by the researchers, all using a standardized protocol. 

Genetic, demographic, pre-operative, and post-operative clinical data were compiled. The movement and disability subscales of the Burke Fahn Marsden Dystonia Rating Scale (BFM-M/D) were used to assess patients before and after surgery [27,28]. The maximal BFM movement (BFM-M) score is 120 and the maximal disability (BFM-D) score 30, with a lower score indicating less severe dystonia [27]. Baseline and follow-up raw scores (BFM-M/D) are presented individually (Table 1, Figure 1), as median and interquartile range (IQR) in the result section, and as boxplots with whiskers indication minimum to maximum in Figure 2. Changes from baseline in BFM-M/D scores were calculated and are reported as percentages. A reduction of 25% or more in the BFM movement score was used to identify DBS responders [18,29]. For each individual, the time-point after surgery when the maximal DBS effect had been reached was identified and the corresponding BFM movement or disability score recorded. This maximal effect reflects the initial response to the procedure. The score from the last recorded follow-up is also reported and reflects how stable this effect was over time. For some individuals with a short follow-up, the maximal effect and last follow-up scores are identical.

Statistical analyses were performed using the SPSS version 25 software (IBM, Armonk, NY, USA). Wilcoxon signed rank test was used for ordinal data to analyze potential differences in dystonia burden assessed by BFM M/D before and after GPi-DBS. Effect size was calculated after having used the Wilcoxon signed rank test using the formula *r* = z/√n, where *z* = the result obtained from the test statistics, and *n* = number of observations. An effect size of *r* = 0.1 was considered small, *r* = 0.3 medium, and *r* = 0.5 or more large. The Spearman correlation coefficient was used to explore correlations between different variables (age at surgery, disease duration before surgery, preoperative disease burden) and outcome (percent change in BFM-M at the time for maximal effect compared to baseline). The size of the correlation coefficient was interpreted as negligible (0.0 < 0.3), low (0.3 < 0.5), moderate (0.5 < 0.7), or high (0.7–0.9).

The study was pre-approved by the Regional Ethics Committee in Stockholm, Sweden, (no 2017/983-31/1). Informed consent was obtained from all participants and, when appropriate, their caregivers. The study adhered to the recommendations of the Helsinki declaration.

## 3. Results

### 3.1. Baseline Characteristics

Fourteen individuals, 9 females and 5 males, with dystonia and sequence variants in *THAP1* were included. Median age at dystonia onset was 9 years (range 4–42 years). Several oral medications and/or botulinum toxin injections had been administered to all patients before DBS, with no noteworthy or lasting clinical improvement in any case. Baseline characteristics are shown in Table 1. For patient number 13, baseline BFM-M/D scores are missing because the initial surgical procedure was performed in another center. The scores in Table 1 for this patient are before and 1 year after a revision of the DBS-system, performed 3 years after the initial surgery.

### 3.2. Timing of Surgery and Follow-Up 

Surgery was performed after a median disease duration of 9 years (range 2–19 years) (Table 2). At the time of DBS surgery, all individuals had generalized or segmental dystonia. All 14 had oro-laryngeal dystonia, causing speech impairment, and 5 were very severely affected or even anarthric. In 6 patients, to various extent, swallowing was affected. Three had documented musculoskeletal contractures prior to DBS, in the neck (patient 6 and 7) and in the feet (patient 12). The follow-up procedure differed between the 5 contributing centers, as did the length of follow-up for each patient. The median follow-up time after DBS was 4 years and 10 months, range 7 months–16 years. 

### 3.3. Clinical Outcome

A beneficial effect of DBS surgery was reported for all 14 individuals (Table 2, Figure 1). Overall, a clinically relevant improvement became evident rapidly, in 5 patients within 1 month and in 6, within 3 months. For patient 8, the favorable effect appeared 6 months after DBS. For 2 patients (1 and 11), the time when a clinically relevant improvement was first evident was not known. The maximal beneficial effect was reached after a median of 10 months (range 6–24 months). In most patients, the improvement was stable during the subsequent follow-up period, but in 4 patients, the dystonia worsened somewhat over the years (Figure 1 and Figure 2). A video of patient 9 is available as Appendix A as an illustration of the clinical outcome (3 preoperative and 3 postoperative sequences). 

The BFM-M score was reduced from a median of 52.5 (IQR 35.8–62.0) before surgery to 18.5 (IQR 14.4–37.3) at the last follow-up (*p* = 0.001; effect size *r* = 0.62) at a median time of 4 years and 10 months. The BFM-D score showed a similar effect with a reduction from a median of 10.0 (IQR 8.3–14.5) before DBS to 7.0 (IQR 5.3–9.5) at the last follow-up (*p* = 0.006; effect size *r* = 0.52). An effect size >0.5 was considered large, indicating that the improvement at last follow-up was large as assessed by both the BFM-M (effect size 0.62) and BFM-D (effect size 0.52).

As per the definition of a DBS-responder (see methods), 12 of our 14 patients could be classified as responders (Table 2). However, patient 3 was later reclassified as a secondary non-responder, with worsening of symptoms and only 21% reduction of dystonia 11 years after DBS compared to baseline as assessed by the BFM-M. For the whole group, the median reduction in the BFM-M score was 63% (IQR 45–77) when maximal effect was reached and 58% (IQR 31–62) at the last follow-up. The corresponding median reductions in the BFM-D score were 45% (IQR 3–56) and 32% (IQR 15–46). 

The overall improvement in dystonia following DBS was greater in the trunk and limbs than in the cranio-cervical and oro-laryngeal regions. In 10/14 patients, there was no detectable effect on speech or swallowing. The remaining 4 patients (2, 3, 12, and 13) demonstrated improvement of speech or swallowing, but less obvious than that in other regions affected by dystonia.

### 3.4. Correlation Timing of Surgery and Clinical Outcome

We could not identify any significant correlation between age at surgery (Spearman correlation coefficient 0.04, *p* = 0.905), disease duration before surgery (Spearman correlation coefficient 0.28, *p* = 0.336), or disease burden at the time for surgery (Spearman correlation coefficient 0.20, *p* = 0.502) and the effect of DBS, as assessed by the percent change in BFM-M score at the time for maximal effect compared to baseline. The individuals with contractures preoperatively (patient 6, 7, and 12) were all responders to the DBS, with 68%, 78%, and 46% reductions of dystonia at last follow-up, respectively, as assessed by the BFM-M score (Table 2).

### 3.5. Complications and Side Effects

DBS-device related complications and side-effects of DBS therapy were infrequent. Dysphonia related to stimulation parameters was reported in two patients. In one individual, an idiopathic edema along the lead tracks, later successfully treated with corticosteroids, was reported. Patients 11 and 13 underwent intracerebral revisions (Table 2). Patient 11 received a second pair of pallidal electrodes, since a progressive worsening of the cranial and upper-limb dystonia could not be sufficiently controlled with the initial DBS leads. The second surgery led to further improvement in the motor assessment, suggesting a possible somatotopic effect dependent on the location of the leads. In patient 13, the contact between electrodes and battery was disrupted and when this was discovered and corrected, 3 years after the original implant, an abdominal wound infection developed that required prolonged antibiotic treatment. This patient subsequently improved, but presented 3 years later, 6.3 years after the primary implant, with rapid loss of speech and difficulty swallowing, requiring a replacement of one intracerebral electrode. After the replacement, effective swallowing, normal oral feeding, and quiet understandable speech returned, although this patient failed to achieve the >25% reduction in the BFM movement score required to qualify as a responder.

## 4. Discussion

This study, representing the largest assessment of the efficacy of DBS in patients with DYT-THAP1 dystonia to date, included 14 individuals treated with GPi-DBS and a median follow-up of about 5 years. Our data confirm that GPi-DBS is an effective treatment option for these patients, with a median reduction of 58% in the BFM-M score at the last follow-up. Additionally, the maximal effect was observed early, after a median time of 10 months and, most importantly, the improvement remained stable throughout the period of follow-up in the majority of patients. However, DBS failed to improve the confining oro-laryngeal dystonia in more than 70% of the subjects enrolled. In this limited population with DYT-THAP1 dystonia, no obvious differences between responders and non-responders, were identified that predicted clinical outcome after pallidal DBS.

Bilateral GPi-DBS is documented as a safe and effective treatment option for patients with severe generalized and segmental dystonia [8,10]. The variation of the response to the treatment in different dystonias is not fully understood. It is obvious that at least some of the variation can be explained by genetic etiology [19]. Examples of monogenic dystonias known to improve to different degrees after pallidal DBS include DYT-TOR1A dystonia (also known as DYT1 dystonia), DYT-KMT2B dystonia, and DYT-ATP1A3 dystonia (also known as DYT12 dystonia). The first dystonia for which the specific genetic cause was identified was DYT-TOR1A dystonia and the comprehensive data concerning DBS on such patients reveals excellent and long-lasting motor improvement [15,16,18,30,31]. In one study including 47 individuals with DYT-TOR1A dystonia, dystonia severity was reduced by an average of about 80% two years after DBS and in some patients, the effect remained stable during follow-up for up to 8 years [30]. In the case of DYT-KMT2B dystonia, Meyer et al. reported that 10 individuals responded well to DBS [32]. However, no quantification of the improvement was reported, thus no comparison to other genetically-determined dystonias is possible. In contrast, DYT-ATP1A3 dystonia (rapid-onset dystonia parkinsonism), caused by mutations in the *ATP1A3* gene, is an example of a combined dystonia disorder where afflicted individuals appear to receive no benefit from DBS, although to our knowledge, only 5 such patients have been reported on so far [33,34]. 

Regarding the effect of DBS on patients with DYT-THAP1 dystonia, available evidence is limited. In the first two reports, published in 2010 and including a total of 6 patients, the outcome was described as less obvious than in the case of DYT-TOR1A dystonia [23,24]. However, a recent study including 9 patients with DYT-TOR1A dystonia, 8 with DYT-THAP1 dystonia, and 38 with unknown forms of isolated dystonia, with a post-operative follow-up between 22 and 92 months, showed similar long-term improvement of BFM-M in DYT-TOR1A and DYT-THAP1 mutation carriers (−44% and −42% respectively) [18]. In another study, three family members, all carrying the same mutation in the *THAP1* gene, were monitored up to 11 years after DBS and two displayed a highly effective response, with 39% and 67% improvement, as assessed by the BFM scale [26]. Our present study supports a good clinical outcome after GPi-DBS in DYT-THAP1 dystonia, with a 58% median reduction in BFM-M score during a median follow-up for about 5 years. It has been proposed that due to a more variable and less predictable outcome after DBS in DYT-THAP1 dystonia than DYT-TOR1A dystonia, genetic testing should be performed preoperatively [18]. Since the etiology of dystonia influences outcome after DBS, we agree that comprehensive genetic testing should be performed preoperatively. According to our findings, carriers of *THAP1*-mutations would then be considered well-suited for surgery. 

Phenotypical differences provide another explanation for the variance in response to DBS in different kinds of dystonia. The anatomical distribution of the dystonia is one important aspect in this connection. Oropharyngeal and cranio-cervical symptoms are often considered to be the most disabling symptoms of DYT-THAP1 dystonia. Unfortunately, the effect of DBS on speech and swallowing in such patients has been disappointing [23,24,26], with only a few individuals demonstrating effective improvement [35,36]. A poor response of dysarthria/anarthria to GPi-DBS is probably not specific to DYT-THAP1 dystonia, since similar findings have been reported in connection with DYT-KMT2B dystonia [32]. One might speculate that orolaryngeal dystonia might benefit from a different localization of the electrodes than in the GPi, although this has not to our knowledge been systematically evaluated.

In our study, the effects on speech and swallowing were very limited, thereby confirming most other reports. Specifically, all 14 individuals had orolaryngeal dystonia before surgery and only 4/14 patients showed some improvement in these regions after DBS. This information can help make expectations more realistic during pre-DBS counselling of patients with DYT-THAP1 dystonia. The response of individuals with DYT-TOR1A to DBS has traditionally been considered superior to that of those with DYT-THAP1. In DYT-TOR1A dystonia, speech and swallowing are typically not affected, which might explain, at least in part, this difference in outcomes.

Several reports and reviews indicate that possible positive predictors, that might explain some of the variations in outcome after DBS include younger age at surgery, shorter disease duration preoperatively, higher disease burden at surgery, and absence of fixed postures [8,12,37,38,39,40]. In contrast to those studies, we did not find any significant correlation between age at surgery, disease duration before surgery, or preoperative disease burden, and outcome of DBS, as assessed by the BFM-M at last follow-up compared to baseline. Moreover, our three patients with fixed postures responded well to DBS. The limited number of patients though, does not allow definitive conclusions. Nevertheless, these data support offering DBS to all pharmaco-resistant DYT-THAP1 patients, regardless of age, disease duration, or presence of fixed deformities. However, the rapid benefits of DBS emphasize the need to consider early surgery in children, to prevent the unnecessary morbidity and disability following years of living with dystonia.

Limitations of the present study include the limited number of patients enrolled, which is related to the rarity of the disorder, the absence of a control group, and the inclusion of patients from different European centers with varying follow-up programs. Furthermore, the pathogenicity of the THAP1 variants has not been confirmed for all sequence changes and some of the changes may represent rather benign variants. However, lack of affected family members on whom to perform segregation and functional assays hampers assessment of the pathogenicity of the novel variants. Another important limitation is that the BFM was the only outcome measure consistently used, whereas important outcomes addressing participation and activity or the perceptions of the individual and their families were not reported in a way that allowed presenting. Finally, the GPi was the only surgical target in our series, but combined pallidal and subthalamic nucleus stimulation has previously been suggested as possibly more efficient than pallidal stimulation only [18].

## 5. Conclusions

In conclusion, this study contributes to the understanding of the long-term motor and disability outcome of DBS in patients with DYT-THAP1 dystonia and, consequently, some clinical recommendations can be made. In DYT-THAP1 dystonia, DBS is a safe and effective procedure both in the short and long term perspective. Its effect is rapidly observed, within the first weeks after surgery, and maximal after a median of 10 months. Therefore, DBS should be considered in patients with DYT-THAP1 dystonia with an unsatisfactory response to drugs. Patients and caregivers should be informed that the improvement after DBS can be variable and that orolaryngeal dystonia is likely to show negligible improvement after surgery, with only approximately 30% of patients achieving some functional relief. In contrast to previous studies, our data suggests that surgery should be offered also to older patients with a long disease course and even to patients with fixed deformities. Nevertheless, DBS should be considered early in children to prevent the blight of dystonia interfering with childhood social, emotional and educational development, and independence. Lastly, we suggest performing comprehensive genetic testing of all patients with dystonia considered potential candidates for DBS, since the underlying molecular defect might contribute significantly to predicting the efficacy and functional outcome of DBS.

## Figures and Tables

**Figure 1 jcm-08-02163-f001:**
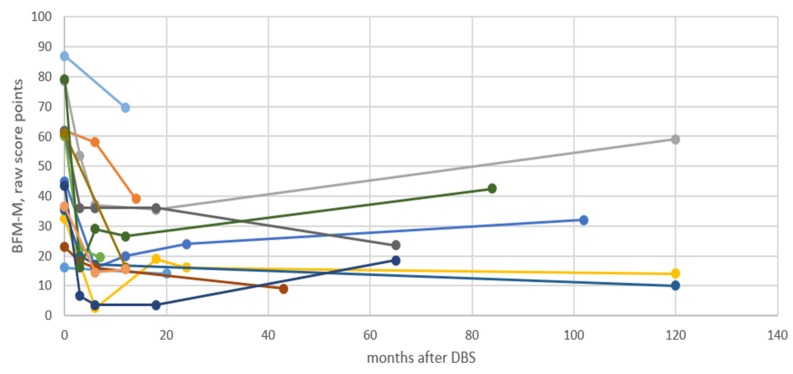
Fourteen individuals with DYT-THAP1 dystonia, followed longitudinally after pallidal deep brain stimulation and evaluated by BFM-M. Note that only 10 years of follow-up is included in the figure. BFM-M = Burke Fahn Marsden movement subscale. DBS = deep brain stimulation.

**Figure 2 jcm-08-02163-f002:**
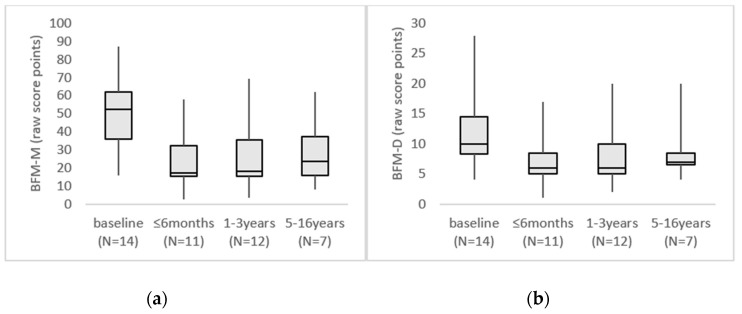
Boxplots showing BFM-M/D raw score points, with whiskers from minimum to maximum score. *N* = the number of individuals for whom data were available at each time-interval. (**a**) BFM-M boxplots at baseline and during follow-up at three different time intervals. (**b**) BFM-D boxplots at baseline and during follow-up at three different time intervals. BFM-M = Burke Fahn Marsden movement subscale, BFM-D = Burke Fahn Marsden disability subscale.

**Table 1 jcm-08-02163-t001:** Baseline characteristics of our 14 subjects with DYT-THAP1 dystonia.

Patient Number	Gender	*THAP1* (NM_018105.3) Sequence Variant	Patient Previously Reported	Variant Previously Reported	Age at Disease Onset (Years)	Initial Anatomical Distribution	Preop BFM-M	Preop BFM-D	Preoperative Medication
**1**	F	c.-34G > A variant in the 5’ untranslated region	no	no	42	cervical	16	4	botulinum toxin, benzodiazepines, gabapentin, NSAID
**2**	F	c.377_378delCT p.Pro126Argfs*2 exon 3	no	Blanchard 2011 PMID: 21520283	9	cervical	62	10	L-dopa, botulinum toxin
**3**	M	c.173T > C p.Phe58Ser exon 2	no	Miyamoto 2014 PMID: 24227593	6	left foot	78.5	28	baclofen, L-dopa, trihexyphenidyl
**4**	F	c.70_71+8del10 p.Gly24fs*71 exon 1	no	no	15	right upper limb	32.5	9	trihexyphenidyl, baclofen
**5**	M	c.464A > C p.Gln155Pro exon 3	no	no	6	right upper limb	45	5	trihexyphenidyl, baclofen
**6**	M	c.238A > G p.Ile80Val exon 2	no	Ledoux 2012 PMID: 22377579 Lohmann 2012 PMID: 21847143 Golanska 2015 PMID: 26087139	14	cervical	60	15	trihexyphenidyl, valproic acid, gabapentin
**7**	F	c.94C > T p.Leu32Phe exon 2	no	no	7	right upper limb	35.5	12	trihexyphenidyl
**8**	F	c.70_71 + 8del10 p.Gly24fs*71 exon 1	no	no	40	cervical	23	6	botulinum toxin, benzodiazepine
**9**	F	c.207_209delCAA p.Asn69-Asn69del exon 2	no	Groen 2010 PMID: 20687191 Clot 2011 PMID: 21110056	6	right upper limb	62	10	L-dopa, trihexyphenidyl
**10**	F	c.85C > T premature stop codon at amino acid position 29 exon 2	no	Djamarti 2009 PMID: 19345148 Bressman 2009 PMID: 19345147 Xiromerisiou 2012 PMID: 22903657 Dobričić 2013 PMID: 23180184	9	right lower limb	61	10	L-dopa, trihexyphenidyl, botulinum toxin
**11**	F	c.16T > C p.Ser6Pro exon 1	Cif 2012 PMID: 22339165 (prior to the DYT-THAP1 diagnosis)	Clot 2011 PMID: 21110056	9	speech	43.5	13	benzodiazepines, trihexyphenidyl, carbamazepine
**12**	M	c.77C > G p.Pro26Arg exon 2	Lumsden 2012 PMID: 23452222 (prior to the DYT-THAP1 diagnosis)	Houlden 2010 PMID: 20211909 Campagne 2012 PMID: 22844099	4	Hands	79	26	L-dopa, trihexiphenidyl, bensodiazepine
**13**	M	c.19G > A p.Ala7Thr exon 1	no	no	6	left foot	87	23	L-dopa
**14**	F	c.98G > A p.Cys33Tyr exon2 variant of uncertain significance	no	no	9	left foot	36.5	8	trihexyphenidyl, L-dopa

F = female; M = male; * = stop codon; BFM-M = Burke Fahn Marsden Dystonia Rating Scale—movement subscore; BFM-D = Burke Fahn Marsden Dystonia Rating Scale—disability subscore; NSAID = non-steroid anti-inflammatory drug; L-dopa = levodopa.

**Table 2 jcm-08-02163-t002:** Outcome after pallidal deep brain stimulation of individuals with DYT-THAP1 dystonia.

Patient Number	Age at GPi-DBS (Years)	Disease Duration Before GPi-DBS (Years)	Length of Follow-Up after GPi-DBS (Years, Months)	Change in BFM-M, Last Follow-Up Compared to Baseline (%)	Responder (>25% Improvement BFM-M)	Effect on Speech and/or Swallowing	Intra-Cerebral Revision (number)	DBS Device at Last Follow-Up (Number of Changes)	Stimulation Frequency at Last Follow-Up (Hz)
**1**	57	15	1 year 8 months	−13	no	no	0	Medtronic Activa RC (1)	130
**2**	14	5	1 year 2 months	−37	yes	some	0	Medtronic Activa PC	130
**3**	13	7	11 years 1 month	−21	initially	some	0	Medtronic Activa RC (2)	130
**4**	17	2	10 years 0 month	−57	yes	no	0	Activa RC (2)	180
**5**	14	8	8 years 7 months	−29	yes	no	0	Medtronic Activa RC (1)	180
**6**	32	18	7 months	−68	yes	no	0	Medtronic Activa SC	100
**7**	26	19	13 years 9 months	−78	yes	no	0	Medtronic Activa SC (1)	RGPi 110; LGPi 90
**8**	54	14	3 years 7 months	−61	yes	no	0	Medtronic Activa SC	RGPi 125; LGPi 90
**9**	11	5	6 years 0 month	−62	yes	no	0	Medtronic Activa RC	130
**10**	22	13	1 year 0 month	−74	yes	no	0	Soletra, medtronic leads	130
**11**	20	11	16 years 4 months	−60	yes	no	1	Medtronic Activa RC	130
**12**	12	8	6 years 11 months	−46	yes	some	0	Activa RC (1)	130
**13**	8	2	1 year 1 month	−20	no	some	2	Medtronic Activa RC (1)	NR
**14**	19	10	1 year 0 month	−58	yes	no	0	Vercise-DBS-system/Boston Scientific	130

GPi-DBS = pallidal deep brain stimulation; BFM-M = Burke Fahn Marsden Dystonia Rating Scale—movement subscore; DBS = deep brain stimulation; RGPi = right globus pallidus interna; LGPi = left globus pallidus interna; (x) = number of changes of DBS device; NR = not reported.

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
