# Peer review of "Pallidal Deep Brain Stimulation in DYT6 Dystonia: Clinical Outcome and Predictive Factors for Motor Improvement"

_jcm, 2019, doi:10.3390/jcm8122163_

Round 1
Reviewer 1 Report
Pallidal deep brain stimulation (DBS) is an established treatment in dystonia. In this paper, Dr. Annika Danielsson and co-workers try to elucidate the clinical outcome of GPi-DBS on individuals with DYT-THAP1 dystonia and to identify potential factors that predict positive or negative changes in motor function. By creatively utilization of the Burke-Fahn-Marsden Dystonia Rating Scale method, they found that DBS is a safe and effective procedure in DYT-THAP1 dystonia both in the short and long term perspective. They also suggested that pallidal deep brain stimulation should be considered in clinically impairing and pharmaco-resistant DYT-THAP1 dystonia. The style and overall representation of the article are satisfying; the study methods are properly described with accuracy and are suitable for J. Clin. Med. Furthermore, the authors need to make a minor correction before this manuscript is finally accepted for publication.
Particular comments:The reference cited needs to check with great cautious since some information is incomplete. For example, the published year or page numbers for Refs. 13, 17, 37 are missing. The senior author needs proofread of the revision with great caution.
Reviewer 2 Report
This multi-center study adds significantly to the available evidence regarding long-term DBS outcomes in generalized dystonia for DYT6 patients. Limited prior evidence suggests that these patients benefit less from DBS than DYT1 patients. This study further supports that observation, though provides evidence that benefit from DBS for DYT6 patients may still be substantial and long-lasting.
Author Response
Thank you very much for nice comments!
Kind regards Annika Danielsson
Reviewer 3 Report
Danielsson et al. evaluated the clinical outcome of pallidal DBS in DYT-THAP1 dystonia in 14 patients. This is a very important study that could significantly enhance our understanding of the long-term motor and disability outcome of DBS in patients with DYT-THAP1 dystonia, and also can help with evaluating the safety and effectiveness of DBS procedure in dystonia treatment. This manuscript is well written with a good standard of English. The manuscript is worthy of consideration for publication though a few questions need to be addressed first.
Have the authors evaluated and analysed the clinical outcome from each centre separately? I think the authors should discuss this and include it in the manuscript, results from such an analysis might provide more details to improve the procedure effectiveness. In page 1, line 38, remove the extra ). In page 3, line 113, please define z and n in the formula.Author Response
Please see the attachment.
